# Is the Leptin/Adiponectin Ratio a Better Diagnostic Biomarker for Insulin Resistance than Leptin or Adiponectin Alone in Adolescents?

**DOI:** 10.3390/children9081193

**Published:** 2022-08-09

**Authors:** César Agostinis-Sobrinho, Sofia E. de Castro Ferreira Vicente, Sigute Norkiene, Alona Rauckienė-Michaelsson, Justina Kievisienė, Viney Prakash Dubey, Arturas Razbadauskas, Luís Lopes, Rute Santos

**Affiliations:** 1Faculty of Health and Sciences, Klaipeda University, 92294 Klaipeda, Lithuania; 2Research Centre in Physical Activity, Health and Leisure, Faculty of Sport, University of Porto, 4200-450 Porto, Portugal; 3Laboratory for Integrative and Translational Research in Population Health, University of Porto, 4200-450 Porto, Portugal

**Keywords:** leptin, adiponectin, fatness, youth

## Abstract

Background: Recently, the leptin/adiponectin (L/A) ratio has been suggested as a novel predictor of cardio-metabolic and other chronic diseases. Aim: To evaluate the ability of leptin (L), adiponectin (A), and the L/A ratio in identifying high risk of insulin resistance IR in adolescents, adjusted by cardiorespiratory fitness, adherence to the Mediterranean diet, and body fat percentage. Subjects and methods: This is a cross-sectional analysis with 529 adolescents aged 12–18 years-old. Blood samples were taken to analyze glucose, insulin, leptin, and adiponectin levels. IR (homeostasis model assessment of insulin resistance (HOMA-IR) was estimated from fasting serum insulin and glucose). Results: Adiponectin, leptin, and L/A ratio were accurate to predict IR among adolescents. The optimal L/A cut-off value to indicate risk of IR development was >0.35 in boys and >0.97 in girls. Logistic analyses showed that the suggested cut-off points for adiponectin (girls: OR: 2.87 (1.26–6.53); *p* = 0.012); leptin (boys: OR: 5.23 (1.16–7.14) *p* = 0.006; girls: OR: 2.99 (1.10–8.09) *p* = 0.031), and the L/A ratio (boys: OR: 8.38 (2.6–26.8) *p* < 0.001; girls: OR: 6.1 (2.1–17.0) *p* < 0.001), were significant predictors of IR, after adjustments for age, pubertal stage, adherence to the Mediterranean diet, cardiorespiratory fitness, and body fat percentage. Conclusion: Leptin and L/A ratio were associated with IR risk, after adjustments for confounders in both sexes and adiponectin in girls. The L/A ratio seems to have a higher diagnostic accuracy to identify IR risk than adiponectin or leptin, in both sexes.

## 1. Introduction

Adipocytokines, such as leptin and adiponectin, are adipose-tissue-derived molecules and have been associated with insulin resistance (IR), obesity, metabolic syndrome, atherogenesis, and acute cardiovascular events, leading to several metabolic outcomes [1].

Leptin is a multifunctional metabolic regulator implicated in food intake, energy expenditure, and neuroendocrine function. Leptin has been found to be an important mediator of the obesity-related pro-inflammatory state that contributes to metabolic disorders [2,3]. However, hyperleptinemia, a common condition among the obese, has been associated with IR and endothelial dysfunction disorders, which suggests an overall decrease in sensitivity to leptin [4]. Indeed, a recent meta-analysis (5) reported a significant association between leptin and the risk of cardiovascular diseases (CVD) and strokes. On the other hand, hypoadiponectinemia has been proposed to be an independent CVD risk factor [5], associated with increased cardiovascular events and mortality rates [5]. Therefore, the mechanisms associated with the role of adiponectin as an anti-inflammatory adipokine includes its key action on insulin-sensitizing, anti-inflammatory, and anti-atherogenic effects [6].

Recently, the interest in the leptin/adiponectin (L/A) ratio as a novel predictor of cardio-metabolic and other chronic disease outcomes has increased [7]. This ratio has been associated with IR [8], carotid intima-media thickness [9], and metabolic syndrome [7,10], among others. Conversely, the assessment of the L/A ratio has been proposed as a useful parameter to assess IR predisposition and as a possible atherosclerotic index for patients with type 2 diabetes [10].

Interestingly, most of the knowledge about the relationship between adipocytokines and IR is derived from studies focusing on adult obese subjects or adults with metabolic syndrome [11]. However, it is known that childhood diabetes carries a higher risk of several adult metabolic diseases [12]. Therefore, these risks make childhood diabetes an important avenue of investigation, especially considering the rising number of obese children and adolescents in not only developed, but also developing countries [13].

For instance, some studies have shown that leptin and adiponectin are associated with IR [14,15]. However, these associations were not always independent of other potential confounders, such as dietary intake, body fatness, and cardiorespiratory fitness. For example, cardiorespiratory fitness [16], body fatness [17], and adherence to a Mediterranean diet [18,19] have been shown to have key roles in insulin resistance. Additionally, none of the previous studies in children and adolescents have analyzed the L/A ratio’s ability to predict IR.

In this context, the present study aimed, (i) to evaluate the ability of leptin, adiponectin, and the L/A ratio to predict IR, (ii) and the association between these hormones with IR, independently of adherence to a Mediterranean dietary pattern, body fatness, and cardiorespiratory fitness in adolescents.

## 2. Materials and Methods

### 2.1. Study Design and Sample

The adolescents included in this analysis were part of the Longitudinal Analysis of Biomarkers and Environmental Determinants of Physical activity (LabMed Physical Activity Study), which is a school-based prospective cohort study carried out in four cities in northern Portugal. We selected 5 secondary schools, which had already collaboration agreements established with our research center and, therefore, they were selected primarily for pragmatic, budgetary, and logistical reasons. Participants’ recruitment was conducted at the selected schools, and all pupils belonging to the 7th and 10th grade classes were invited to participate in the study (*n* = 1678). The LabMed Physical Activity Study methodologies have been presented elsewhere [20]. From that initial sample, 534 adolescents agreed to undergo blood sampling. However, 5 of them were later excluded from the analysis due to C-reactive protein values >10 mg/L, which may be indicative of acute inflammation or illness, leaving 529 adolescents (267 girls, mean age 14.3 ± 1.7 years) as the final sample for the present study. For the present study, power analysis was calculated post hoc (for α = 0.05), and it was higher than 0.8 for ROC curves and Logistic Regression Analysis sample size calculation in both sexes.

### 2.2. Ethical and Legal Requirements

The study was conducted in accordance with the World Medical Association’s Helsinki Declaration for Human Studies. The Portuguese Data Protection Authority (#1112434/2011), the Portuguese Ministry of Science and Education (0246200001/2011), and the Ethics Committee of the Faculty of Sport, University of Porto, approved the study. All participants in this study were informed of the study’s goals, and written informed consent was obtained from participating adolescents and their parents or guardians. Considering potential refusals to participate in the study due to blood analysis, a “layered consent” was permitted, allowing the participants to consent to some parts of the study protocol and not others. For example, an adolescent could perform physical fitness assessments and refuse to undergo blood sampling. Throughout the study, no exclusion criteria were applied to avoid discrimination. However, for the present analysis, only apparently healthy adolescents were considered, i.e., without any medical diagnosis of physical or mental impairment.

### 2.3. Measures

#### 2.3.1. Anthropometrics

Body height was measured to the nearest 0.1 cm in bare or stockinged feet, with the adolescent standing upright against a portable stadiometer (Seca 213, Hamburg, Germany). Body weight was measured to the nearest 0.10 kg, with the participant lightly dressed using a portable electronic weight scale (Tanita Inner Scan BC 532, Tokyo, Japan). BMI was calculated as weight divided to height squared (kg/m^2^), and the participants were classified as underweight, normal weight, overweight or obese using the age and sex-specific cut-off values proposed by the International Obesity Task Force. To assess the body fat percentage, a frequency current of 50 kHz (Tanita Inner Scan BC 532, Tokyo, Japan) was used to measure leg-to-leg Bio impedance. After manual introduction of the sex, age and height into the scale system, participants were asked to come up and remain still on the scale until measurement was completed, fulfilling the manufacturer’s instructions.

#### 2.3.2. Pubertal Stage

Participants self-assessed their pubertal stage of secondary sex characteristics (breast and pubic hair development for girls, genital and pubic hair development for boys; ranging from stage I to V), according to the criteria of Tanner and Whitehouse [21].

#### 2.3.3. Blood Sampling

Blood samples were obtained from each subject early in the morning, following a 10 h overnight fast by venipuncture from the antecubital vein. The samples were stored in sterile blood collection tubes in refrigerated conditions (4°C to 8 °C) for no longer than 4 h during the morning of collection and then sent to an analytical laboratory for testing according to standardized procedures, as follows: (i) serum leptin and serum adiponectin by plate reader method (ELISA analyzer); (ii) glucose by hexokinase method (Siemens Advia 1600/1800 Erlangen, Germany); (iii) insulin by chemiluminescence immunoassay (Siemens ACS Centaur System, Erlangen, Germany. All assays were performed in duplicate according to the manufacturers’ instructions. Insulin resistance homeostatic model assessment (HOMA-IR) was calculated as the product of basal glucose (mmol/L) and insulin (µL U/mL) levels divided by 22.5, and was used as a proxy measure of insulin resistance (23). HOMA-IR were transformed to standardized values (Z-score = (participant’s value—mean value of the sample)/standard deviation)) by age and sex. High risk (at risk) was considered when the individual had ≥1 SD (standard deviation) of this Z-score, as previously suggested [22].

#### 2.3.4. Adherence to the Mediterranean Diet

To assess the degree of adherence to the Mediterranean diet, the KIDMED index (Mediterranean Diet Quality Index for children and adolescents) was used [23] (25). The index is based on 16-questions, self-administered, which sustain the principles of the Mediterranean dietary patterns, as well as those that undermine it. The results of the index varied between 0 and 12 points. The questions that have one negative connotation in relation to Mediterranean diet were equal to (−1); the questions that constitute positive aspects were equal to (+1). A continuum variable was computed to perform the statistical analyses.

#### 2.3.5. Cardiorespiratory Fitness

Cardiorespiratory fitness was measured using the 20-metre shuttle run test. This test requires participants to run back and forth between two lines set 20 m apart. Running speed started at 8.5 km/h and increased by 0.5 km/h each minute, reaching 18.0 km/h at minute 20. Each level was announced on a tape player. The participants were instructed to keep up with the pace until exhausted. The test was finished when the participant failed to reach the end lines concurrent with the audio signals on two consecutive occasions. Otherwise, the test ended when the subject stopped because of fatigue. The participants received verbal encouragements from the investigators to achieve maximum performance, to keep running as long as possible. The number of laps performed by each participant was recorded [24].

#### 2.3.6. Statistical Analysis

Measurement data mean and standard deviations were evaluated for normality by the Kolmogorov–Smirnov test and compared using the t-test to examine sex differences.

A receiver-operating-characteristic curve (ROC) was drawn to calculate area under the curve (AUC), and the best cut-off value of the leptin, adiponectin, and L/A ratio to discriminate between low/high values of IR were determined using the maximum Youden’s index. Youden’s index is defined as: J = (sensitivity + specificity – 1); the critical threshold value is the point at which the sensitivity and specificity is maximized. The positive likelihood ratio LR (+) and the negative likelihood ratio LR (−) were used to analyze the potential diagnostic accuracy. The prognostic accuracy of serum leptin, adiponectin, and L/A ratio for predicting insulin resistance was performed by comparison of area under curve (AUC) of ROC curves. The AUC of ROC curves was derived by the Hanley and McNeil method [25].

Based on the cut-off values suggested by the ROC curves, logistic regression analyses were performed to further study the relationships between adiponectin, leptin, and L/A ratio and HOMA-IR, adjusted for the following potential confounders: age, pubertal stage, cardiorespiratory fitness, adherence to a Mediterranean diet, and body fat percentage.

Data analysis was performed using the Statistical Package for the Social Sciences for Windows (Version 21.0 SPSS Inc., Chicago, IL, USA) and MedCalc (Version 11.1.1.0 MedCalc, Mariakerke, Belgium). A *p* value < 0.05 denoted statistical significance.

## 3. Results

Participant characteristics are presented in Table 1. Girls presented higher levels of adiponectin, leptin, and body fat percentage (*p* < 0.001 for all), whereas boys presented high cardiorespiratory fitness levels (*p* < 0.001).

Table 2 shows the parameters of the ROC curves analysis for the diagnostic performance of adiponectin, leptin, and L/A ratio in predicting high IR risk in girls and boys. ROC curve analyses revealed that adiponectin, leptin, and the L/A ratio were accurate (*p* < 0001) to predict high IR risk, with the highest AUC values achieved by L/A ratio in both girls (0.736 CI–95% 0.679–0.788 *p* < 0.001) and boys (0.823 CI 95% 0.771–0.867 *p* < 0.001). For boys, the L/A ratio and leptin ROC curves were significantly different from adiponectin (*p* < 0.05), whereas for girls, the L/A ratio ROC curve was significantly different from adiponectin and leptin (*p* < 0.05).

Logistic analyses showed that adiponectin in girls (Table 3) (OR: 2.87 (1.26–6.53 95% CI); *p* = 0.012), leptin in boys (Table 4) (OR: 5.23 (1.16–17.14) *p* < 0.001) and in girls: (OR: 2.99 (1.10–8.09) *p* = 0.031)), and the L/A ratio in boys (OR: 8.38 (2.62–26.81) *p* < 0.001) and in girls (OR: 6.11 (2.18–17.05) *p* < 0.001), were significant predictors of IR in adolescents, after adjustments for age, pubertal stage, adherence to the Mediterranean Diet, and cardiorespiratory fitness and body fat percentage.

## 4. Discussion

The main findings of this study suggest that (i) adiponectin, leptin, and the L/A ratio present a good discriminatory ability in identifying a high IR risk in adolescents; (ii) the L/A ratio seems to have a higher diagnostic accuracy in identifying a greater IR risk than adiponectin in both sexes; and (iii) leptin and the L/A ratio are associated with high IR risk independent of potential confounders (age, pubertal stage, adherence to the Mediterranean diet, cardiorespiratory fitness, and body fat percentage) in both sexes and adiponectin only in girls.

ROC curves predicting high IR were analyzed by sex, resulting in better sensitivity for boys and higher specificity for girls. Additionally, the adiponectin AUC of ROC curve was statistically different from leptin and the L/A ratio for boys. Because of physiological differences, girls present higher circulating levels of leptin and fatness than boys [26]. In our study, adiponectin is not statistically different between sexes (girls 8.9 vs. boys 9.0 mg/L); however, leptin levels present a large difference between them (boys 1.16 vs. girls 5.4 ng/mL). In our current line of thinking, these results may partially explain the difference between sexes in the AUC of ROC curves.

Previously, it was suggested that the L/A ratio was an important pro-inflammatory biomarker and a powerful independent predictor of intima-media thickness in healthy subjects, and a better predictor than leptin or adiponectin alone [9]. Once the L/A ratio had been proposed as a useful indicator for adipocyte dysfunction [27], in a recent study, we suggested that the L/A ratio may be a clinically significant indicator for predicting a clustering of metabolic risk factors in adolescents [28]. Along the same line, another study found that levels of L/A had a stronger association with metabolic syndrome than either adiponectin or leptin separately, especially when controlled for BMI in Chinese youth [29]. In our study, the L/A ratio, as well as leptin, shows the higher diagnostic accuracy in identifying a greater IR risk. However, when we assess the association of the suggested cut point, the L/A ratio has the highest odds ratio to detect a greater IR risk. Together, these findings have public health and clinical implications, as childhood and adolescence are propitious periods of life to identify those at risk for CVD and to develop strategies to improve their metabolic health status.

It is important to note that this is the first study that compared the ability of the adiponectin, leptin, and L/A ratio in determining IR among adolescents. Although leptin (for boys and girls) and adiponectin (for girls) had statistically significant ROC curves, the best AUCs were presented by the L/A ratio in both sexes (Table 2). In fact, Finucane et al. [30] reported in two large population-based samples, the hypothesis that L/A is strongly associated with the IR and that this ratio was more effective as a diagnostic parameter of IR than leptin and adiponectin analyzed separately. Additionally, other studies that evaluated the L/A ratio considered it a useful parameter for predicting IR in patients with and without diabetes [8,10]. However, a pediatric study, comparing the diagnostic value of the L/A ratio to leptin or adiponectin, is still lacking.

Another important result observed in the present study was that leptin is a positive predictor of IR in adolescents. Corroborating this finding, Aldhoon-Hainerová et al., showed recently that leptin was a significant predictor of HOMA-IR [31]. It is known that leptin has insulin-sensitizing effects [3]. However, high leptin levels may also upregulate proinflammatory cytokines such as TNF-α and IL-6 that contribute to IR, besides inducing the expression of vascular adhesion molecules and the pro-thrombotic tissue factor [32].

According to our findings, we were able to identify a significant relationship between adiponectin and IR risk in girls. Indeed, lower adiponectin concentrations have been found in patients with IR, obesity, type 2 diabetes, and coronary artery disease, compared with control subjects [33,34]. Interestingly, in a study by Vicente et al. [35] that analyzed the adiponectin pathway, it was shown that the adiponectin behavior was partially mediated by a negative correlation with the L/A ratio, HOMA-Adiponectin and HOMA-IR; and by a positive correlation with HDL-cholesterol in the highest tertile of the adiponectin. Adiponectin may be related with atherosclerosis, either directly or indirectly, through improvements in IR. Adiponectin suppresses most of the processes involved in atherosclerotic vascular changes, including the expression of adhesion molecules in vascular endothelial cells, the proliferation of vascular smooth muscle cells, and the formation of foam cells [36].

The strengths of our study include the consideration of important confounding variables in our analysis, such as cardiorespiratory fitness, body fatness, and dietary intake. In fact, food consumption can be a predictor of metabolic health. Previous studies reported that the adherence to a Mediterranean dietary pattern might have a dual effect on the prevention of IR as well as metabolic syndrome, CVD, by improving classical risk factors and by having an intense anti-inflammatory effect [19]. In addition, adolescence is a period of natural changes in several metabolic systems such as body composition and sex hormones [37]. However, in our study, the analyses were controlled for age, pubertal stage, body fatness, and stratified by sexes.

Limitations of this study include its cross-sectional design, which precludes causal attributions; in addition, we gathered dietary intake through the Kidmed questionnaire. Although it is the most widely instrument used to score MedDiet in children and adolescents, further validations regarding the Mediterranean diet are needed.

In conclusion, we showed that adiponectin, leptin, and the L/A ratio presented discriminatory ability in identifying a high IR risk in adolescents. Moreover, the use of the L/A ratio seems to have a higher diagnostic accuracy in identifying an unfavorable IR profile than adiponectin or leptin, independent of potential confounds.

## Figures and Tables

**Table 1 children-09-01193-t001:** Participants’ characteristics.

Characteristics	Mean (SD)		
	All (*n* = 529)	Girls (*n* = 267)	Boys (*n* = 262)
Age (year)	14.33 (±1.73)	14.27 (±1.71)	14.39 (±1.74)
Weight (kg)	55.15 (±12.81)	53.44 (±11.18)	56.89 (±14.10) *
Height (cm)	160.27 (±9.59)	157.7 (±6.68)	162.9 (±11.3) *
BMI (kg/m^2^) IOTF	21.31 ± 3.84	21.41 ± 3.96	21.20 ± 3.73
UW/NW/OW/OB (n)	24/357/111/37	10/181/54/22	14/176/57/15
UW/NW/OW/OB (%)	4.5%/67.5%/21%/7%	3.7%/67.8%/20.3%/8.2%	5.3%/67.2%/21.8%/5.7%
Leptin (ng/mL)	4.12 (± 4.93)	6.21 (± 5.6)	1.98 (± 2.85) *
Ratio Leptin/Adiponectin	0.43 (±0.67)	0.62 (± 0.81)	0.23 (± 0.39)
Insulin resistance (HOMA-IR)	3.45 (±5.38)	3.51 (±1.83)	3.39 (±7.41)
KIDMED Index	7.11 (±2.05)	7.19 (±1.98)	7.02 (±2.13)
20 m shuttle run test (Nr. of laps)	44.90 (±1.12)	31.87 (±15.25)	58.15 (±26.63) *

* Significantly different from girls, *p* < 0.001, Independent Two-tailed *t*-Tests. Abbreviations: BMI, body mass index (according to the age and sex-specific cut-off values of the International Obesity Task Force (IOTF)): UW, underweight; NW, normal weight; OW, overweight; OB, obese; SD, standard deviation.

**Table 2 children-09-01193-t002:** Parameters of the ROC curves analysis for the diagnostic performance of adiponectin, leptin and L/A ratio in predicting high risk of insulin resistance in girls and boys.

		AUC	95% CI	*p* Value	Cut-Off	Sensitivity (%)	Specificity (%)
Girls (*n* = 267)	Adiponectin (mg/L)	0.670	0.640–0.753	0.0005	≤8.9	54.3	76.3
	Leptin (ng/mL)	0.700	0.640–0.753	0.0001	>5.4	77.1	63.4
	L/A ratio *	0.736	0.679–0.788	0.0001	>0.97	54.3	89.2
Boys (*n* = 262)	Adiponectin (mg/L)	0.603	0.541–0.663	0.0640	≤9.0	67.6	52.6
	Leptin (mg/L) **	0.806	0.753–0.852	0.0001	>1.16	85.3	62.7
	L/A ratio **	0.823	0.771–0.867	0.0001	>0.35	67.6	86.8

AUC, Area under the curve; CI, Confidence intervals; Positive likelihood ratio LR (+); Negative likelihood ratio LR (−); L/A, leptin/adiponectin ratio. * Significantly different from AUC for adiponectin and leptin in girls (*p* < 0.05), ** Significantly different from AUC for adiponectin in Boys (*p* < 0.05).

**Table 3 children-09-01193-t003:** Number and percentage of girls below and above the cut-off value and associations between leptin, adiponectin and LA ratio, and risk group of insulin resistance.

HOMA-IR	**Cut-Off**	***n* (%)**		**OR**	**95% CI**	** *p* **
	Adiponectin					
	≥8.9 mg/L	193 (72.3)		1		
	<8.9 mg/L	74 (27.7)	Model 1	3.92	(1.86–8.27)	<0.001
			Model 2	2.87	(1.26–6.53)	0.012
	Leptin					
	<5.4 ng/mL	155 (58.1)		1		
	≥5.4 ng/mL	112 (41.9)	Model 1	6.18	(2.61–14.66)	<0.001
			Model 2	2.99	(1.10–8.09)	0.031
	Ratio LA					
	>0.97	222 (83.1)		1		
	≥0.97	45 (16.9)	Model 1	10.71	(4.62–24.82)	<0.001
			Model 2	6.11	(2.18–17.05)	<0.001

Model 1: adjusted for age, pubertal stage; Model 2: Model 1 additionally adjusted for adherence to the Mediterranean diet, cardiorespiratory fitness, and body fat percentage.

**Table 4 children-09-01193-t004:** Number and percentage of boys below and above the cut-off value and associations between leptin, adiponectin and LA ratio, and risk group of insulin resistance.

HOMA-IR	**Cut-Off**	***n* (%)**		**OR**	**95% CI**	** *p* **
	Adiponectin					
	≥9.0 mg/L	131(50.0)		1		
	<9.0 mg/L	131(50.0)	Model 1	2.40	(1.10–5.17)	0.026
			Model 2	2.08	(0.88–4.89)	0.092
	Leptin					
	<1.16 ng/mL	148(56.5)		1		
	≥1.16 ng/mL	114(43.5)	Model 1	10.11	(3.73–27.41)	<0.001
			Model 2	5.23	(1.16–17.14)	0.006
	Ratio LA					
	<−0.35	207(79.1)		1		
	≥−0.35	55(21.0)	Model 1	13.61	(5.93–31.24)	<0.001
			Model 2	8.38	(2.62–26.81)	<0.001

OR, odds ratios; Model 1: Adjusted for age, pubertal stage; Model 2: Model 1 additionally adjusted for adherence to the Kidmed cardiorespiratory fitness and body fat percentage.

## Data Availability

Not applicable.

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
