# Peer review of "Is the Leptin/Adiponectin Ratio a Better Diagnostic Biomarker for Insulin Resistance than Leptin or Adiponectin Alone in Adolescents?"

_children, 2022, doi:10.3390/children9081193_

Round 1
Reviewer 1 Report
It is a beautiful and significant study highlighting the use of LA ratio as an important biomarker for predicting IR in adolescents.
Title can be improved and needs to be more suggestive. may be say - Leptin/Adiponectin ratio as a better biomarker for prediction of IR in adolescents.
Abstract, introduction and methods are fine.
In results section- there are table-3 and 4. There is no description for those tables and also they are not cited anywhere in the paper.
Discussion is well written.
Author Response
Reviewer #1:
It is a beautiful and significant study highlighting the use of LA ratio as an important biomarker for predicting IR in adolescents.
Authors: Thank you for the positive feedback regarding our study, as well as the constructive input.
Title can be improved and needs to be more suggestive. may be say - Leptin/Adiponectin ratio as a better biomarker for prediction of IR in adolescents.
Authors: Thank you. We changed it for: “Is the leptin/adiponectin ratio a better diagnostic biomarker for insulin resistance than leptin or adiponectin alone in adolescents?”
Abstract, introduction and methods are fine.
Authors: Thank you
In results section- there are table-3 and 4. There is no description for those tables and also they are not cited anywhere in the paper.
Authors: Thank you for point this out. This was emended.
Discussion is well written.
Authors: Thank you
Reviewer 2 Report
The authors present a cross-sectional study on the predictive power of leptin, adiponectin and L/A ratio for insulin resistance in children under consideration of major confounders.
The overall rationale of the paper is clear and is introduced orderly and transparently.
The methods are mainly well described. Please specify the medical exclusion criteria, as some diagnosis might have been accepted (dyslipoproteinemia, allergies, ...), while others were certainly deemed worth excluding (severe diseases of internal organs...).
Results: Presented very clearly; robust statistics; overall plausible findings.
Table 1: Asterisks of different size still have the same meaning? Please clarify.
Discussion: No need for changes.
Excellent work!
Author Response
Reviewer #2:
The authors present a cross-sectional study on the predictive power of leptin, adiponectin and L/A ratio for insulin resistance in children under consideration of major confounders.The overall rationale of the paper is clear and is introduced orderly and transparently.
Authors: Thank you for the positive feedback.
The methods are mainly well described. Please specify the medical exclusion criteria, as some diagnosis might have been accepted (dyslipoproteinemia, allergies, ...), while others were certainly deemed worth excluding (severe diseases of internal organs...).
Authors: Done. Thank you
Results: Presented very clearly; robust statistics; overall plausible findings.
Authors: Thank you
Table 1: Asterisks of different size still have the same meaning? Please clarify.
Authors: This was a formatting error. We amended. Thank you
Discussion: No need for changes.
Authors: Thank you
Excellent work!
Authors: Thank you